# Identification of *NUDT15* gene variants in Amazonian Amerindians and admixed individuals from northern Brazil

**Juliana Carla Gomes Rodrigues[1], Tatiane Piedade de Souza[1], Lucas Favacho Pastana[1], André Maurício Ribeiro dos Santos[2], Marianne Rodrigues Fernandes[1], Pablo Pinto[1,2], Alayde Vieira Wanderley[3], Sandro José de Souza[4], José Eduardo Kroll[4], Adenilson Leão Pereira[2], Leandro Magalhães[2], Laís Reis das Mercês[2], Amanda Ferreira Vidal[2], Tatiana Vinasco-Sandoval[2], Giovanna Chaves Cavalcante[2], João Farias Guerreiro[2], Paulo Pimentel de Assumpção[1], Ândrea Ribeiro-dos-Santos[1,2], Sidney Santos[1,2], Ney Pereira Carneiro dos Santos[1,2]\***

1 Núcleo de Pesquisas em Oncologia, Universidade Federal do Pará, Belém, Pará, Brazil, 2 Laboratório de Genética Humana e Médica, Instituto de Ciências Biológicas, Universidade Federal do Pará, Belém, Pará, Brazil, 3 Hospital Ophir Loyola, Departamento de Pediatria, Belém, Pará, Brazil, 4 Brain Institute, Universidade Federal do Rio Grande do Norte, Natal, Rio Grande do Norte, Brazil

\* npcsantos.ufpa@gmail.com

**Data Availability Statement:** All relevant data are within the paper and its Supporting Information files.

## Abstract

### Introduction

The nudix hydrolase 15 (*NUDT15*) gene acts in the metabolism of thiopurine, by catabolizing its active metabolite thioguanosine triphosphate into its inactivated form, thioguanosine monophosphate. The frequency of alternative *NUDT15* alleles, in particular those that cause a drastic loss of gene function, varies widely among geographically distinct populations. In the general population of northern Brazilian, high toxicity rates (65%) have been recorded in patients treated with the standard protocol for acute lymphoblastic leukemia, which involves thiopurine-based drugs. The present study characterized the molecular profile of the coding region of the *NUDT15* gene in two groups, non-admixed Amerindians and admixed individuals from the Amazon region of northern Brazil.

### Methods

The entire *NUDT15* gene was sequenced in 64 Amerindians from 12 Amazonian groups and 82 admixed individuals from northern Brazil. The DNA was extracted using phenol-chloroform. The exome libraries were prepared using the Nextera Rapid Capture Exome (Illumina) and SureSelect Human All Exon V6 (Agilent) kits. The allelic variants were annotated in the ViVa® (Viewer of Variants) software.

### Results

Four *NUDT15* variants were identified: rs374594155, rs1272632214, rs147390019, andrs116855232. The variants rs1272632214 and rs116855232 were in complete linkage disequilibrium, and were assigned to the *NUDT15\*2* genotype. These variants had high

**Funding:** We acknowledge funding from Conselho Nacional de Desenvolvimento Científico e Tecnológico- CNPq (http://www.cnpq.br) through the call Chamada Universal- MCTI / CNPq No. 14/2013, contemplated researcher Paulo Pimentel de Assumpção; Universidade Federal do Pará- UFPA (https://portal.ufpa.br); Pró-Reitoria de Pesquisa e Pós-Graduação da UFPA- PROPESP (http://www.propesp.ufpa.br); Coordenação de Aperfeiçoamento de Pessoal de Nível Superior-CAPES(https://www.capes.gov.br); Fundação Amazônia de Amparo a Estudos e Pesquisas-FAPESPA (http://www.fapespa.pa.gov.br). The funders had no role in study design, data collection and analysis, decision to publish, or preparation of the manuscript.

**Competing interests:** The authors have declared that no competing interests exist.

frequencies in both our study populations in comparison with other populations catalogued in the 1000 Genomes database. We also identified the *NUDT15*4* haplotype in our study populations, at frequencies similar to those reported in other populations from around the world.

## Conclusion

Our findings indicate that Amerindian and admixed populations from northern Brazil have high frequencies of the *NUDT15* haplotypes that alter the metabolism profile of thiopurines.

## Introduction

The analysis of the polymorphisms of the Drug Absorption, Distribution, Metabolism, and Excretion (ADME) genes has been commonly employed in personalized medicine as a diagnostic tool for the selection of the most appropriate drugs and/or dosage for the treatment of a range of diseases. A number of recent studies [1, 2, 3] have identified varying frequencies in many genetic polymorphisms in genetically distinct populations, which may modulate the prognosis of patients and the therapeutic efficacy of treatments.

One ADME-related gene that represents a major breakthrough in the field of pharmacogenetics is the nudix hydrolase 15 (*NUDT15*) gene. This gene acts in the metabolism of thiopurine, catabolizing its active metabolite thioguanosine triphosphate (TGTP) into its inactivated form, thioguanosine monophosphate (TGMP) [4]. Studies have shown that genetic variants responsible for the loss of *NUDT15* function lead to an increased incorporation of thioguanine nucleotides into the DNA strand, which would result in an exacerbation of the associated cytotoxic effects, culminating in the appearance of hematological toxicities, such as severe myelosuppression [5, 6].

Thiopurine-based drugs, such as 6-mercaptopurine (6-MP) and azathioprine (AZA), are widely used to treat Acute Lymphoblastic Leukemia (ALL) and Inflammatory Bowel Disease (IBD), respectively. However, the efficacy or tolerance of these drugs may vary considerably according to the level of activity of the NUDT15 enzyme in the patient [7, 8]. The American Food and Drug Administration (FDA) recently recommended specific dose adjustments of both 6-MP and AZA according to the *NUDT15* genotype profile of the patient [9, 10].

The NUDT15 gene show six main haplotypes (*1-*6), composed by the four variants combinations (p.Val18_Val19insGlyVal, p.Val18Ile, p.Arg139Cys, p.Arg139His), as described by Moriyama and colleagues [11]. Also, *NUDT15* still has assigned eight actionable dyplotypes, which are sorted by enzymatic activity level: normal (*1/*1), intermediate (*1/*2,*1/*3,*1/*4,*1/*5) and low (*3/*5,*2/*3,*3/*3) [11, 12].

Data available from international databases (the 1000 Genomes Project, the Exome Aggregation Consortium [ExAC], and the Genome Aggregation Database [gnomAD]) show that the frequencies of alternative *NUDT15* alleles, especially those that cause a drastic loss of gene function, vary greatly between geographically distinct populations. A C>T transition in the rs116855232 (p.Arg139Cys) variant is assigned to two haplotypes (*NUDT15*2* and *NUDT15*3*), which present an extreme loss of gene function, associated with the presence of the T allele of this polymorphism. The T allele of rs116855232 is found more commonly in East Asian [13], Latin [14] and Native American [15] populations, but is rare in populations from Europe, Africa, and South Asia.

The *NUDT15*9 haplotype is derived from an in-frame deletion in the sequence responsible for the catalytic activity of NUDT15 (rs746071566), and is associated with extreme loss of enzyme function. This deletion has been found exclusively in patients of European or African descent [16].

An in-frame insertion in this same region (rs1272632214; p.Gly17_Val18dup) is assigned to two haplotypes (*NUDT15*2 and *NUDT15*6), which are both related to a drastic loss of gene function. This insertion has been identified in populations from Guatemala, Singapore, and Japan, and enzymatic assays have shown that this insertion is responsible for a reduction in the catalytic activity of the NUDT15 enzyme [11].

In Brazil, thiopurine-based therapy is part of the standard treatment protocol for ALL. In a recent study developed by our research group on a northern Brazilian population, severe toxicities (grades 3 and 4) were found in 65% of the patients treated with the standard therapeutic protocol for ALL [7]. This toxicity rate is much higher than that observed in other populations (worldwide average = 26%) treated using the same protocol [7, 17].

Previous data demonstrated that Brazilian Native American populations have a high frequency of a polymorphism (rs116855232) that drastically reduces NUDT15 activity, and that these ethnic group contribute 30%, on average, of the genetic makeup of the admixed populations of the Brazilian Amazon region [15, 18, 19]. Thereafter, a possible cause of the severe toxic effects observed in northern Brazil during the standard thiopurine-based treatment protocol is that individuals of this region have higher frequencies of *NUDT15* deleterious alleles, resulting from the genomic contribution inherited of Amerindian groups. Our objective is to evaluate the frequency of deleterious alleles of *NUDT15* gene in Amazonian Amerindians and admixed populations of northern Brazil.

Therefore, we employed next-generation sequencing (NGs) data obtained by our research group to define the molecular profile of the *NUDT15* gene in two samples, one containing 64 individuals from 11 different indigenous groups of the Brazilian Amazon region and the other, 82 individuals representing the admixed population from the same region.

## Methods

### Study population

The study population is composed of 64 Amerindians and 82 admixed individuals from the Amazon region of northern Brazil. The Amerindians represent 12 different Amazonian ethnic groups that were grouped together as the Native American (NAM) group. Details such as name, location and number of individuals in each ethnic group are described in S1 Table. The 82 admixed individuals live in Belém, city located in northern Brazil, and, due to its colonization process, are characterized of, mainly, three ancestral genetic components: European, Native American and African. These admixed individuals were enrolled in a broader project to investigate germline mutations in patients with gastric cancer, and are referred to in the present study as the Brazilian Admixed Population (BAP). The present study was approved by the National Committee for Ethics in Research (CONEP) and the Research Ethics Committee of the UFPA Tropical Medicine Center, under CAAE number 20654313.6.0000.5172. All participants signed a free-informed consent as well as the tribe leaders and when necessary a translator explained the project and the importance. The recruitment period for participants was from September 2017 to December 2018.

We compared our results with those of populations from other continents obtained from the phase 3 release of the 1000 Genomes database (available at http://www.1000genomes.org). These population include those of African (AFR), Latin (AMR), East Asian (EAS), European (EUR), and South Asian (SAS) descent. We also compared our findings with genomic data

from two databases of genomic variants analyzed in Brazilian populations, specifically individuals from southeast Brazil: the Online Archive of Brazilian Mutations (ABraOM) (available at http://abraom.ib.usp.br) and the Brazilian Initiative on Precision Medicine (BIPMed) (available at https://bipmed.org/).

### Extraction of the DNA and preparation of the exome library

The DNA was extracted using the phenol-chloroform method described by Sambrook et al. [20]. The genetic material was quantified using a Nanodrop-8000 spectrophotometer (Thermo Fisher Scientific Inc., Wilmington, DE, USA) and the integrity of the DNA was assessed by electrophoresis in 2% agarose gel.

The libraries were prepared using the Nextera Rapid Capture Exome (Illumina) and Sure-Select Human All Exon V6 (Agilent) kits following the manufacturer's recommendations. The sequencing reactions were run in the NextSeq 500® platform (Illumina®, US) using the Next-Seq 500 High-output v2 300 cycle kit (Illumina®).

### Bioinformatic analysis

The quality of the FASTQ reads was analyzed (FastQC v.0.11- http://www.bioinformatics. babraham.ac.uk/projects/fastqc/), and the samples were filtered to eliminate low-quality readings (fastx_tools v.0.13—http://hannonlab.cshl.edu/fastx_toolkit/). The sequences were mapped and aligned with the reference genome (GRCh38) using the BWA v.0.7 tool (http://bio-bwa.sourceforge.net/). Following this alignment with the reference genome, the file was indexed and sorted (SAMtools v.1.2—http://sourceforge.net/projects/samtools/). Subsequently, the alignment was processed for duplicate PCR removal (Picard Tools v.1.129—http://broadinstitute.github.io/picard/), mapping quality recalibration, and local realignment (GATK v.3.2—https://www.broadinstitute.org/gatk/). The results were processed to determine the variants from the reference genome (GATK v.3.2). The analysis of the variant annotations was run in the ViVa® (Viewer of Variants) software, developed by the Federal University of Rio Grande do Norte (UFRN) bioinformatics team. The databases and their versions used for variant annotations are: SnpEff v.4.3.T, Ensembl Variant Effect Predictor (Ensembl release 99) and ClinVar (v.2018-10). For *in silico* prediction of pathogenicity, it was used: SIFT (v.6.2.1), PolyPhen-2 (v.2.2), LRT (November, 2009), Mutation Assessor (v.3.0), Mutation Taster (v. 2.0), FATHMM(v.2.3), PROVEAN(v.1.1.3), MetaSVM(v1.0), M-CAP(v1.4) e FATHMM-MKL (http://fathmm.biocompute.org.uk/about.html).

### Statistical analyses

The allele frequencies of the populations were obtained directly by gene counting, and compared with the other study populations (AFR, EUR, AMR, EAS, SAS, ABraOM and BIPMed). The difference in frequencies between the populations were analyzed by Fisher's exact test, results were considered significant when $p$-value $\leq 0.05$. The inter-population variability of the polymorphisms was assessed using Wright's fixation index (FST). We performed a linkage disequilibrium estimation using Haploview v. 4.2. The analyses were run in Arlequin v.3.5 [21] and in RStudio v.3.5.1.

### Results

We identified four *NUDT15* variants, two INDELs (Insertion/Deletion) and two SNVs (Single Nucleotide Variant) in the 146 individuals analyzed, with a medium coverage of 77X. Table 1 provides details of the characteristics of these variants, including their chromosomal location

**Table 1. Description of the *NUDT15* variants found in the 146 individuals sampled in the present study.**

| SNP or INDEL position | Reference SNP Id | Location | Impact predicted by SNPeff | Change in AA | ClinVar significance | *In silico* prediction of pathogenicity[a] |
|---|---|---|---|---|---|---|
| 13:48037782 | rs1272632214 | Exon 1 | Moderate | p. Gly17_Val18dup | - | - |
| 13:48045720 | rs147390019 | Exon 3 | Moderate | p.Arg139His | Drug response to thiopurines | Yes (LRT/Mutation Taster) |
| 13:48045719 | rs116855232 | Exon 3 | Moderate | p.Arg139Cys | Drug response to thiopurines | Yes (LRT) |
| 13:48037953 | rs374594155 | Intron 1 | Modified | - | - | - |

[a]The other *in silico* analysis tools used did not show pathogenic impact.

and genic region, the reference number, ClinVar significance, the *in silico* prediction of functional impact, and pathogenicity.

The rs374594155 variant corresponds to a deletion of 25 nucleotide pairs located in intron 1 of the *NUDT15* gene. No published information is available on the possible phenotypic or functional effects of this mutation.

Two variants (rs147390019 and rs116855232) provoke changes in the amino acid sequence. Both have clinical significance for the response of the organism to thiopurine-based drugs, as indicated by the ClinVar. *In silico* predictions of pathogenicity have also been reported. The rs1272632214 variant is derived from the insertion of six nucleotide pairs into exon 1, which results in the in-frame insertion of two amino acids (Glycine and Valine) into the catalytic activity region of the enzyme.

The Hardy-Weinberg Equilibrium (HWE) was calculated and $p$-values of less than 0.05 were considered significant. All the polymorphisms reported were in HWE, except for rs374594155 ($p$-value < 0.001).

The allele frequencies recorded for the four variants in the two study populations are relatively high, in general, in comparison with the five populations obtained from the 1000 Genome database and the samples of Brazilian populations from the ABraOM and BIPMed databases, as shown in Table 2.

The in-frame insertion of six nucleotide pairs (rs1272632214) has an allele frequency of 9.4% in the Amerindian population analyzed here, and a frequency of 6.8% in the admixed population. These frequencies are slightly higher than those recorded in the AMR (5%) and EAS (6%) populations, and much higher than those observed in the EUR (1%), ABraOM (0,1%) and BIPMed (0,5%) populations. The in-frame mutation is absent from the AFR and SAS populations.

**Table 2. Comparison of the allele frequencies recorded for the *NUDT15* variants in the two populations analyzed in the present study (NAM and BAP) with those of five continental populations (AFR, AMR, EAS, EUR and SAS) described in the 1000 genomes database.**

| Reference SNP Id | Minor Allele Frequency | | | | | | | | |
|---|---|---|---|---|---|---|---|---|---|
| | NAM | BAP | AFR | AMR | EAS | EUR | SAS | ABraOM | BPIMed |
| rs1272632214 | 0.0938 | 0.0680 | 0 | 0.050 | 0.060 | 0.010 | 0 | 0.001 | 0.005 |
| rs147390019 | 0.0160 | 0.0067 | 0 | 0.007 | 0.001 | 0 | 0 | 0.004 | - |
| rs116855232 | 0.0938 | 0.0680 | 0.001 | 0.045 | 0.095 | 0.002 | 0.070 | 0.012 | - |
| rs374594155[a] | 0.0550 | 0.1300 | 0.087 | 0.120 | 0.080 | 0.008 | 0.062 | 0.060 | - |

[a]Intron variant

The C>T transition that defines the rs116855232 was identified only in the individuals that presented the above-mentioned in-frame polymorphism, both in the Amerindian population (9.4%) and in the Amazonian admixed population (6.8%). In both populations, the rs1272632214 and rs116855232 are in complete linkage disequilibrium (D = 1 and $R^2$ = 1). This variant (T allele) is more frequent in Asian (EAS = 9.5% and SAS = 7%) and Latin (AMR = 4.5%) populations, but is very rare in Africans (AFR = 0.1%), Europeans (EUR = 0.2%) and in southeast Brazilian (ABraOM = 1,2% and absent in BIPMed).

The G>A transversion that defines the rs147390019 was identified in two Amerindians (frequency of 1.6%) and one individual from the admixed population (0.6%). This variant is considered rare (frequency less than 1%) in all other populations from around the world.

The Table 3 shows the pairwise comparisons of frequencies found between the Amerindians and the admixed Brazilian population of this study, the five populations of the 1000 genomes project and the sample of Brazilians from the ABraOM database.

The results demonstrated that the rs116855232 variant has the most discrepant frequencies between the Amerindians and admixed Brazilian populations in comparison with the others; each showed significant differences with Africans, Europeans and Brazilians from ABraOM. Indeed, when we analyze the pairwise differences by the prism of parental population, the European is the most distinctive regarding both Amerindians and admixed Brazilian population from Amazon (Table 3).

The Table 4 reports the inter-populational variability of the three exonic variants found in the Amazonian Amerindian and the admixed Brazilian population, which was assessed by the Wright's fixation index (FST). The most significant differences were reported between the NAM and AFR groups (0.47507), followed by the NAM and EUR groups (0. 39153) and, the NAM e ABraOM populations (0. 29119).

**Table 3. Pairwise comparison of allelic frequencies in Amerindians and admixed population of Brazil with each one of the five continental populations from 1000 genomes and the southeast population of the ABraOM databases.**

| Pairwise Comparison (P-Value[a]) | Reference SNP Id | | | |
|---|---|---|---|---|
| | rs1272632214 | rs147390019 | rs116855232 | rs374594155[b] |
| NAM vs. BAP | 1 | 1 | 1 | 1 |
| NAM vs. AFR | - | - | **1.358e-05** | 1 |
| NAM vs. AMR | 0.5708 | 1 | 0.7583 | 1 |
| NAM vs. EAS | 1 | 0.6385 | 1 | 1 |
| NAM vs. EUR | **0.0018** | - | **5.858e-05** | **0.0439** |
| NAM vs. SAS | - | - | 1 | 1 |
| NAM vs. ABraOM | **1.409e-05** | 0.7251 | **2.207e-03** | 1 |
| BAP vs. AFR | - | - | **6.831e-05** | 1 |
| BAP vs. AMR | 1 | 1 | 1 | 1 |
| BAP vs. EAS | 1 | 1 | 1 | 0.9482 |
| BAP vs. EUR | **0.0082** | - | **2.807e-04** | **1.3203e-06** |
| BAP vs. SAS | - | - | 1 | 0.2347 |
| BAP vs. ABraOM | **7.496e-05** | 0.8936 | **0.0143** | 0.1267 |

[a]Calculated by Fisher's Exact Test, p-values adjusted by Bonferroni's method.

[b]Intronic variant.

The Brazilian population from BIPMed database was not included since only one allele of one variant was observed.

**Table 4. Pairwise FST among Amerindians, admixed Brazilian population, the five continental populations from 1000 genomes database and the Brazilian population from ABraOM.**

|  | NAM | AFR | AMR | EAS | EUR | SAS | ABRAOM | BAP |
|---|---|---|---|---|---|---|---|---|
| NAM | 0.00000 | | | | | | | |
| AFR | 0.47507 | 0.00000 | | | | | | |
| AMR | 0.05655 | 0.04984 | 0.00000 | | | | | |
| EAS | 0.04913 | 0.08537 | 0.01038 | 0.00000 | | | | |
| EUR | 0.39153 | -0.00034 | 0.03826 | 0.07116 | 0.00000 | | | |
| SAS | 0.12447 | 0.07298 | 0.01209 | 0.01250 | 0.05958 | 0.00000 | | |
| ABRAOM | 0.29119 | 0.00655 | 0.02847 | 0.06181 | 0.00409 | 0.04099 | 0.00000 | |
| BAP | 0.03128 | 0.10400 | -0.00323 | 0.01100 | 0.06932 | 0.01380 | 0.03603 | 0.00000 |

*The individuals from BIPMed database were not included in the test since only one allele of one variant was identified in this database.

## Discussion

The present study is the first to investigate the complete sequence of the *NUDT15* gene in Amazonian Amerindians and a highly admixed population with a major Amerindian component, groups that are under-represented, in general, in pharmacogenetic studies.

We described four variants of the *NUDT15* gene, one of which (rs374594155) is intronic, while the other three (rs147390019, rs1272632214, and rs116855232) are all exonic. These variants all have considerable allelic frequencies and well-established clinical impacts in terms of the tolerance of thiopurine-based treatment protocols and the development of severe toxicity. Two of the variants (rs1272632214, and rs116855232) are in complete linkage disequilibrium.

Based on the Pharmacogene Variation Consortium (PharmVar) classification, only two variant haplotypes were observed in our samples [21]. One (*NUDT15*4*) is represented by the isolated variant rs147390019. The other (*NUDT15*2*) is defined by the combined presence of rs1272632214, and rs116855232.

More than 20 allelic variants described in the *NUDT15* gene are capable of reducing or even deactivating the enzymatic activity of the protein and provoking cytotoxic effects [22, 23, 24]. However, most of these variants are rare and unique to a specific population or group.

The international population databases show that the frequency of the mutations that reduce the functional activity of the enzyme, such as those described by the PharmVar Consortium, exceeds 1% only in Asians, Latin American and Amerindian populations [14, 15, 25]. The most frequent mutations of these groups include the rs116855232 variant, either in isolation or combined with rs1272632214.

The rs147390019 and rs1272632214 polymorphisms were first associated with hematopoietic toxicity in a cohort of children diagnosed with ALL from Japan, Guatemala, and Singapore [11]. In this study, the functional nucleotide phosphatase activity of the NUDT15 enzyme was also described. The evidence indicates that the presence of rs147390019 and rs1272632214 may result in the loss of 75% and 85% *in vitro* enzyme function, respectively. The study [11] also confirmed the absence of the enzyme catabolizing function when the rs1272632214 and rs116855232 variants were combined. Overall, then, the three variants identified here modify substantially the enzymatic function of NUDT15 and can have a significant impact on the response of the carriers of this genotype to thiopurine-based treatments.

Our analysis showed that 22% of the Amerindians investigated and 15% of the admixed individuals have haplotypes with known clinical impacts. The *NUDT15*2* haplotype is more frequent in both populations (NAM = 9.4%, BAP = 6.8%) than *NUDT15*4* (NAM = 1.6%, BAP = 0.7%).

There is one specific investigation that evaluated the presence of rs116855232 of the *NUDT15* gene in Amerindian populations from southern and midwestern Brazil [15]. In this study, the mean frequency of rs116855232 was 25%, ranging from 5% to 32%, generally higher than the frequency recorded in the present study (9.4%). The differences observed among different Amerindian ethnicities may be due to the stochastic effects of genetic drift, which are common in Brazilian Amerindian groups [26, 27, 28]. In fact, we observed that the frequencies of variants in each of the Amerindians tribes, in which the number of individuals is equal or above five, have great variability (S2 Table). For example, the *NUDT15*2* has a frequency of 3% in the Asurini de Trocará group whereas in the Zo'é population is 30%; thus, this reinforce the existence of allelic fluctuations in different Amerindians tribes, wich are arising from evolutionary processes, particularly genetic drift.

In the case of the admixed population, we compared our data with those available in two public genomic databases from the state of São Paulo, in southeastern Brazil: BIPMed, which comprises 106 individuals; and ABraOM, comprising 609 individuals. In the BIPMed database, it was reported 11 polymorphisms in the *NUDT15* gene, however most of them had a frequency lower than 1%. In regard with the variants reported in this study, only one was also described in the database, the rs1272632214, with a very low frequency (only one out of 212 alleles) (http://bipmed.iqm.unicamp.br).

In the ABraOM database, it was possible to identify eight exonic mutations with possible phenotypic effects, of which five were identified in only one (rs746071566, rs149436418, rs115012598, and rs200980982) or two (rs139551410) individuals of the 609 evaluated (http://abraom.ib.usp.br/). The other three mutations were the same as those described in our study, i.e., rs116855232, with a frequency of 1.2%, rs1272632214 (0,1%), and rs147390019 (0.4%) [29].

In summary, the maximum proportion of individuals with deleterious alleles in the population from southeastern Brazil is much lower than the one we recorded in the admixed population from northern Brazil. One possible reason for this discrepancy is the greater contribution of Amerindian ancestry to the northern (Amazonian) population in comparison with that from southeastern Brazil, since the first has an average of 30% and the second has a medium of 8.8% of Amerindian genetic background [17, 18, 30].

The Amerindian ancestry has also been linked to higher frequencies of *NUDT15* deleterious alleles, specifically the rs1272632214 and rs116855232 polymorphisms, in other populations, such as the subpopulations from the Latin population of 1000 genomes database (S2 Table). The Mexicans (MXL), Colombians (CML), Puerto Ricans (PUR), and Peruvians (PEL) and have around of 25%, 25%, 13% [31], and 77% [32] of Amerindian genetic background; and the rs1272632214 and rs116855232 frequencies observed in these populations are, respectively, 2.1% and 2.1% for CML, 3.1% and 4.7% for MXL, 10.6% and 11.8% for PEL and, lastly, 0.5% and 0.5% for PUR, which can be inferred that the populations with higher Amerindian genomic ancestry have higher frequencies of the deleterious alleles, i.e Peruvians, followed by Mexicans.

## Conclusions

Our results demonstrate that Amerindian and admixed populations from northern Brazil have a high frequency of two *NUDT15* haplotypes (*2 and *4) that alter significantly the thiopurine metabolization profile. Our findings indicate that these deleterious mutations could partly account for the high rates of toxicity found in ALL patients form northern Brazil. It will nevertheless be essential to evaluate the practical effects of these haplotypes in case-control studies with patients undergoing 6-mercaptopurine therapy, in order to establish their potential association with the poor clinical outcomes.

## Supporting information

**S1 Table. Name, location and number of individuals in each population group studied.**
(PDF)

**S2 Table. Allelic frequency independently for each of the Amerindian populations (N ≥ 5), of the admixed Brazilian population, of the subpopulations of the American group from the 1000 genomes, and of the southeast Brazilian samples.** CML = Colombians from Medellin, Colombia; MXL: Mexican Ancestry from Los Angeles USA; PEL = Peruvians from Lima, Peru; PUR = Puerto Ricans from Puerto Rico.
(PDF)

**S1 Dataset.**
(PDF)

## Author Contributions

**Conceptualization:** Juliana Carla Gomes Rodrigues.

**Data curation:** Alayde Vieira Wanderley, João Farias Guerreiro.

**Formal analysis:** Juliana Carla Gomes Rodrigues, André Maurício Ribeiro dos Santos, Marianne Rodrigues Fernandes, Pablo Pinto.

**Funding acquisition:** Paulo Pimentel de Assumpção, Ândrea Ribeiro-dos-Santos, Sidney Santos.

**Investigation:** Juliana Carla Gomes Rodrigues, Sidney Santos, Ney Pereira Carneiro dos Santos.

**Methodology:** Adenilson Leão Pereira, Leandro Magalhães, Laís Reis das Mercês, Amanda Ferreira Vidal, Tatiana Vinasco-Sandoval, Giovanna Chaves Cavalcante.

**Project administration:** Ney Pereira Carneiro dos Santos.

**Resources:** Juliana Carla Gomes Rodrigues, Marianne Rodrigues Fernandes.

**Software:** Sandro José de Souza, José Eduardo Kroll.

**Supervision:** Ney Pereira Carneiro dos Santos.

**Validation:** Juliana Carla Gomes Rodrigues, André Maurício Ribeiro dos Santos, Marianne Rodrigues Fernandes, Pablo Pinto.

**Visualization:** Juliana Carla Gomes Rodrigues, André Maurício Ribeiro dos Santos, Marianne Rodrigues Fernandes, Pablo Pinto.

**Writing – original draft:** Juliana Carla Gomes Rodrigues, Tatiane Piedade de Souza, Lucas Favacho Pastana.

**Writing – review & editing:** Juliana Carla Gomes Rodrigues, Sidney Santos, Ney Pereira Carneiro dos Santos.

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
