## [Decision Letter · Decision Letter 0]

12 Dec 2019

PONE-D-19-30849

Whole exome sequencing of Amerindians and admixed individuals from northern Brazil: identification of NUDT15 gene variants

PLOS ONE

Dear Fernandes,

Thank you for submitting your manuscript to PLOS ONE. After careful consideration, we feel that it has merit but does not fully meet PLOS ONE’s publication criteria as it currently stands. Therefore, we invite you to submit a revised version of the manuscript that addresses the points raised during the review process.

Both reviewers have serious objections to what appears to be the main message of the manuscript, namely that NUDT15 gene variants could explain the higher rate of toxicity of thiopurine treatments in NE Brazil, and that those variants are linked to Ameridian ancestry. I agree that you should either redimension the goals and conclusions of your manuscript, or provide the necessary evidence (which implies new analyses) to back up this statement.

We would appreciate receiving your revised manuscript by Jan 26 2020 11:59PM. To enhance the reproducibility of your results, we recommend that if applicable you deposit your laboratory protocols in protocols.io, where a protocol can be assigned its own identifier (DOI) such that it can be cited independently in the future. For instructions see: http://journals.plos.org/plosone/s/submission-guidelines#loc-laboratory-protocols

We look forward to receiving your revised manuscript.

Kind regards,

Francesc Calafell

Academic Editor

PLOS ONE

Journal Requirements:

2. Please provide additional details regarding participant consent. In the ethics statement in the Methods and online submission information, please ensure that you have specified whether consent was written or verbal/oral. If consent was verbal/oral, please specify: 1) whether the ethics committee approved the verbal/oral consent procedure, 2) why written consent could not be obtained, and 3) how verbal/oral consent was recorded. If your study included minors, please state whether you obtained consent from parents or guardians in these cases. If the need for consent was waived by the ethics committee, please include this information.

3. Please include in your Methods section the date ranges over which you recruited participants to this study.

4. To comply with PLOS ONE submission guidelines please deposit your sequencing data in a publicly available repository (you can find a list of repositories in the link here : https://journals.plos.org/plosone/s/data-availability#loc-recommended-repositories)

Reviewers' comments:

Reviewer's Responses to Questions

**Comments to the Author**

1. Is the manuscript technically sound, and do the data support the conclusions?

Reviewer #1: No

Reviewer #2: Partly

2. Has the statistical analysis been performed appropriately and rigorously? 

Reviewer #1: No

Reviewer #2: No

3. Have the authors made all data underlying the findings in their manuscript fully available?

Reviewer #1: No

Reviewer #2: No

4. Is the manuscript presented in an intelligible fashion and written in standard English?

Reviewer #1: No

Reviewer #2: Yes

5. Review Comments to the Author

Reviewer #1: This is an interesting paper, which describes differences in frequencies of variants in NUDT15 gene among Amazonian Amerindians and admixed Brazilian samples, compared with 1000 Genome database. However, the main point of the paper, which is to relate NUDT15 variants with high frequency of toxic effects due to thiopurine treatment in northern Brazilian patients, were not achieve, since the authors only reports minimum allele frequencies, and did not perform any genetic association or similar analysis to compare patients with thiopurine toxic effects and controls, or with patients without thiopurine toxic effects. Below I describe additional Major and minor corrections:

Page 1, line 4: I would change the short title, since it to something like "Investigating NUDT15 gene variants in Amerindians and population from northern Brazil", since we cannot perform a "Whole-Exome" in one gene;

Page 2, line 37: The authors should change this phrase, since only the coding region of NUDT15 gene was sequenced;

Page 4, line 78: The authors should include a brief explanation about NUDT15 diplotypes;

Page 5, lines 98-101: The hypothesis seems confusing to me. Studies have already been shown that NUDT15 genotypes or haplotypes are related to toxic effects from thiopurine-based treatments. However, here the hypothesis is exactly to see whether the link between NUDT15 genotypes and thiopurine toxic effects exists. I think what that the authors mean is whether the frequency of NUDT15 haplotypes is different in this specific population, which would explain the high frequency of severe toxicities. In addition, the authors cited "alternative NUDT15 genotypes", and what are these alternative genotypes? Therefore, I suggest that the authors rewrite this part, in order to better explain the hypothesis;

Page 5, lines 119-121: The authors should briefly explain what ancestry populations this admixed sample contains (Europeans, sub-Saharan Africans and Native Americans? or other additional ancestral one?). It could be achieved by global ancestry estimations (such as ADMIXTURE algorithm), in order to strength the ancestry population information;

Page 6, line 137: In Bioinformatic analysis section, the authors should provide the database version of all database variant annotation used (ClinVar, LRT, Mutation Taster, and others), regardless the variant annotation software used to achieved the database information (Annovar, VEP, VarStation, ViVa), since pathogenicity and clinical information could change depending on the database version used.

Page 7, lines 152-153. The authors should report at least whether these variants were well covered by the exome. It could be included in one column in Table 1;

Table 1: Are the authors analyze other pathogenicity prediction algorithms rather than LRT and Mutation Taster? For example, SIFT, PolyPhen, CADD. If yes, they should be included in Table 1. If not, the authors should analyze the pathogenicity by these other algorithms and included the results in Table 1;

Page 8, lines 167-169: The difference in frequencies among the populations is very interesting; in this case, the authors should include a statistical test in order to guarantee that these differences are not by chance. If the frequency differences are not by chance, it also would be interesting to answer the question “Why are these frequencies are different in this specific population”. It could be due to some evolutionary processes, such as natural selection, genetic drift, or some problems in the data (too much related individuals, high inbreeding). I think the authors should explore these issues to better explain these high frequency (if it is not by chance);

Page 9, line 179: The authors cited linkage disequilibrium estimation, but they did not describe how linkage disequilibrium was estimated in the methods. In addition, important variant informatios is lacking, for example, Hardy-Weinberg disequilibrium reports.

Page 11, lines 223-226: This phrase does not make sense, because the authors cited that no previous information about NUDT15 variation were report, and the next phrase cites one paper which one variant was investigated in Amerindians (ref 13);

Page 11, lines 230-231: The authors also should investigate the NUDT15 variation in the database provided by the Brazilian Initiative on Precision Medicine (BIPMed) (https://bipmed.org/brave/index.html);

Page 11, lines 239-240: The authors should clearly reference this information, since this information could be based on one paper only and could not reflect the southeastern Brazil samples;

Page 12, lines 247-250: The hypothesis rise by the authors is that alternative NUDT15 genotypes could be involved in toxic effects observed in northern Brazilian patients exposed a thiopurine-based treatment. However, the authors did not perform any experiment to clearly link the presence of these NUDT15 haplotypes and levels of thiopurine: they just report the minor allele frequencies in the two groups (which seems that they were not exposed to thiopurine treatment). Therefore, these conclusions did not answer the objective of the paper, and I suggest that the authors design some experiment to answer this hypothesis, or change the hypothesis and the objectives of the paper.

The authors should correct grammatical and typographical errors throughout the manuscript.

Reviewer #2: The authors address an interesting topic, that concerns the population based metabolism of thiopurine, which has important implications for the treatment of several diseases as Acute Lymphoblastic Leukemia or Inflammatory Bowel Disease. Admixed populations from northern Brazil have revealed much higher levels of toxicity than other world wide populations, according to the references cited by the authors.

The authors aim to identify the variants of the NUDT15 gene and assess the frequencies in two populations, a group of different samples belonging to several Native American populations from Northern Brazil and an admixed population from the same area. Moreover, the authors claim correlation of the Native American ancestry proportion in the admixed populations with the variants related to the high toxicity levels after the administration of theopurine based drugs.

I acknowledge the significance of the study. However, I want to highlight few points that the authors should address:

First, the title of the manuscript is confusing. The title claims Whole exome sequencing of Amerindians and admixed individuals from northern Brazil. However, the study only analyses the NUDT15 gene. This is could be misleading. The authors should clarify if they have generated whole exome data. They also should explain why they only analyze this gene and not the rest of the exome. The title should not only be coherent with the text, but both should be consistent with the data provided.

Second. As the authors discuss, it is relevant to know if the pathogenic variants of the admixed populations are related to the Native American ancestry. However, there is a lack of evidence to support this hypothesis and the authors should perform the additional analyses needed for a solid discussion. Ideally, the authors should use the whole exome data to accurately assess how pathogenic variants are correlated with Native American genetic global and local ancestry.

Third, in the interest of personalized medicine, it would be interesting to see if some Native American populations have higher frequencies of the pathogenic alleles than others. Similarly, the frequencies in the American admixed populations can be correlated or not to the average proportions of the Native American ancestry, depending if the Native American genetic sources were different and had different frequencies of the pathogenic alleles.

Therefore, assuming the authors only have access to the data regarding the NUDT15 gene, the authors should address the following analyses.

An extra table for the allele frequencies independently for each of the 11 Native American populations when the population size is five or higher. This table should also analyze independently the four American admixed populations of the 1000 genomes and include the southeastern Brazilian admixed population. The authors should use this information to confirm and discuss more accurately what they say in line 227.

A table reporting the FST between populations regarding all variants of the region analyzed.

If the authors do not have access to the whole exome data and can not compute the average genomic proportions of ancestral components (Native American, European and sub-Saharan African) for the American admixed populations, they should at least report the values found in the literature for these populations (the amazonian Brazilian admixed population, the southeastern Brazilian admixed population and the four American admixed populations) and see if the frequency of the studied variants is correlated with the Native American ancestry as the authors claim in line 241.

Authors must also make minor but important changes:

Introduction

Line 58: change ethnically by genetically

Methods

Line 114: add detailed information of the location of sampling of the admixed and Native American populations

Detail which regions are analyzed and in which chromosome they are found.

Results

Sort tables 1 and 2 by position

Paragraph starting in line 167 should be supported by a statistical test, like the Chi-squared test.

Correct the sentence of line 206 where it is said that Native American populations are descendants of the analyzed East Asian populations. Native American might be more closely related to East Asian populations than to other populations, but this does not mean they are their descendants.

6. PLOS authors have the option to publish the peer review history of their article (what does this mean?). If published, this will include your full peer review and any attached files.

Reviewer #1: No

Reviewer #2: No

---

## [Author Response · Author response to Decision Letter 0]

24 Jan 2020

Reviewer #1: This is an interesting paper, which describes differences in frequencies of variants in NUDT15 gene among Amazonian Amerindians and admixed Brazilian samples, compared with 1000 Genome database. However, the main point of the paper, which is to relate NUDT15 variants with high frequency of toxic effects due to thiopurine treatment in northern Brazilian patients, were not achieve, since the authors only reports minimum allele frequencies, and did not perform any genetic association or similar analysis to compare patients with thiopurine toxic effects and controls, or with patients without thiopurine toxic effects. Below I describe additional Major and minor corrections:

1) Page 1, line 4: I would change the short title, since it to something like "Investigating NUDT15 gene variants in Amerindians and population from northern Brazil", since we cannot perform a "Whole-Exome" in one gene;

Response: We have changed the short title in accordance with this suggestion.

2) Page 2, line 37: The authors should change this phrase, since only the coding region of NUDT15 gene was sequenced;

Response: We have changed the phrase to highlight that we did only the coding region of NUDT15 gene.

3) Page 4, line 78: The authors should include a brief explanation about NUDT15 diplotypes;

Response: The text has been modified accordingly.

4) Page 5, lines 98-101: The hypothesis seems confusing to me. Studies have already been shown that NUDT15 genotypes or haplotypes are related to toxic effects from thiopurine-based treatments. However, here the hypothesis is exactly to see whether the link between NUDT15 genotypes and thiopurine toxic effects exists. I think what that the authors mean is whether the frequency of NUDT15 haplotypes is different in this specific population, which would explain the high frequency of severe toxicities. In addition, the authors cited "alternative NUDT15 genotypes", and what are these alternative genotypes? Therefore, I suggest that the authors rewrite this part, in order to better explain the hypothesis;

Response: We agreed that the manuscript's hypothesis was not very clear and therefore caused confusion about what we actually intended to accomplish. The objective of our paper was to describe the frequency of deleterious alleles of NUDT15 gene in a parental group forming of the actual population from northern Brazil (Native Americans) and in admixed northern population itself. We rewrote this part of the article and hope that the hypothesis has been cleared up. 

5) Page 5, lines 119-121: The authors should briefly explain what ancestry populations this admixed sample contains (Europeans, sub-Saharan Africans and Native Americans? or other additional ancestral one?). It could be achieved by global ancestry estimations (such as ADMIXTURE algorithm), in order to strength the ancestry population information;

Response: The main ancestries populations of the brazilian north population is the European, African and Native Americans. This data is well-stablished about the Brazilian population since it is a consequence of our colonization process. Also, this information has been confirmed by genetic analysis, using a dataset of 61 Ancestry Informative Markers.

Details about the methodology can be achieved in “Santos NP et al. Assessing individual interethnic admixture and population substructure using a 48-insertion-deletion (INSEL) ancestry-informative marker (AIM) panel. Hum Mutat. 2010;31(2):184-90.” and “Ramos et al. Neither self-reported ethnicity nor declared family origin are reliable indicators of genomic ancestry. Genetica. 2016; 144(3):259-65.”)

6) Page 6, line 137: In Bioinformatic analysis section, the authors should provide the database version of all database variant annotation used (ClinVar, LRT, Mutation Taster, and others), regardless the variant annotation software used to achieved the database information (Annovar, VEP, VarStation, ViVa), since pathogenicity and clinical information could change depending on the database version used.

Response: The manuscript has been modified accordingly.

7) Page 7, lines 152-153. The authors should report at least whether these variants were well covered by the exome. It could be included in one column in Table 1;

Response: We have included the information about the exome coverage in the first paragraph of the Results section. 

8) Table 1: Are the authors analyze other pathogenicity prediction algorithms rather than LRT and Mutation Taster? For example, SIFT, PolyPhen, CADD. If yes, they should be included in Table 1. If not, the authors should analyze the pathogenicity by these other algorithms and included the results in Table 1;

Response: All the pathogenicity prediction algorithms used are described in the subsection “Bioinformatic analysis” of the Methods section. We have not included them in the Table 1 because the only algorithms that showed pathogenicity predictions is the LRT and Mutation Taster.

9) Page 8, lines 167-169: The difference in frequencies among the populations is very interesting; in this case, the authors should include a statistical test in order to guarantee that these differences are not by chance. If the frequency differences are not by chance, it also would be interesting to answer the question “Why are these frequencies are different in this specific population”. It could be due to some evolutionary processes, such as natural selection, genetic drift, or some problems in the data (too much related individuals, high inbreeding). I think the authors should explore these issues to better explain these high frequency (if it is not by chance);

Response: As the other reviewer also suggested, we performed a Fisher’s exact test to statistically evaluate the differences in populations frequencies observed. Also, we have explained in the Discussion section that we believe that these differences in frequencies may be due to genetic drift, a commonly genetic event observed in Amerindian populations.

10) Page 9, line 179: The authors cited linkage disequilibrium estimation, but they did not describe how linkage disequilibrium was estimated in the methods. In addition, important variant information is lacking, for example, Hardy-Weinberg disequilibrium reports.

Response: We added a new subsection called “Statistical analysis” in the Methods section to better explain the tests and sofwtwares we have used in our data analyzes. The linkage disequilibrium was estimated using the Haploview software. Additionally, the Hardy-Weinberg disequilibrium reports has been described in the 198-200 lines. 

11) Page 11, lines 223-226: This phrase does not make sense, because the authors cited that no previous information about NUDT15 variation were report, and the next phrase cites one paper which one variant was investigated in Amerindians (ref 13);

Response: The text has been rewritten.

12) Page 11, lines 230-231: The authors also should investigate the NUDT15 variation in the database provided by the Brazilian Initiative on Precision Medicine (BIPMed) (https://bipmed.org/brave/index.html);

Response: We agreed with the suggestion and data from the BIPMED database was included in our analyzes.

13) Page 11, lines 239-240: The authors should clearly reference this information, since this information could be based on one paper only and could not reflect the southeastern Brazil samples;

Response: We have supported this information using additional data about the subpopulation groups of the Latin population of the 1000 genomes database, i.e. Colombians, Mexican descendants, Peruvians and Puerto Ricans. The paragraph comprises the 308-317 lines.

14) Page 12, lines 247-250: The hypothesis rise by the authors is that alternative NUDT15 genotypes could be involved in toxic effects observed in northern Brazilian patients exposed a thiopurine-based treatment. However, the authors did not perform any experiment to clearly link the presence of these NUDT15 haplotypes and levels of thiopurine: they just report the minor allele frequencies in the two groups (which seems that they were not exposed to thiopurine treatment). Therefore, these conclusions did not answer the objective of the paper, and I suggest that the authors design some experiment to answer this hypothesis, or change the hypothesis and the objectives of the paper.

Response: We clarified our hypothesis and hope that it now agrees with our results and conclusions;

15) The authors should correct grammatical and typographical errors throughout the manuscript.

Response: We sent the manuscript to be revised for a native English speaker professional translator, in order to correct typos and language errors across the text (a declaration has been provided and it is attached at the end of this document).

REVIEWER #2: The authors address an interesting topic, that concerns the population based metabolism of thiopurine, which has important implications for the treatment of several diseases as Acute Lymphoblastic Leukemia or Inflammatory Bowel Disease. Admixed populations from northern Brazil have revealed much higher levels of toxicity than other world wide populations, according to the references cited by the authors.

The authors aim to identify the variants of the NUDT15 gene and assess the frequencies in two populations, a group of different samples belonging to several Native American populations from Northern Brazil and an admixed population from the same area. Moreover, the authors claim correlation of the Native American ancestry proportion in the admixed populations with the variants related to the high toxicity levels after the administration of theopurine based drugs.

I acknowledge the significance of the study. However, I want to highlight few points that the authors should address:

• First, the title of the manuscript is confusing. The title claims Whole exome sequencing of Amerindians and admixed individuals from northern Brazil. However, the study only analyses the NUDT15 gene. This is could be misleading. The authors should clarify if they have generated whole exome data. They also should explain why they only analyze this gene and not the rest of the exome. The title should not only be coherent with the text, but both should be consistent with the data provided.

Commentary: We have changed the title of the manuscript. Also, we have analyzed only the NUDT15 gene because we have a particular clinical condition in our region regarding the toxicities arising from thioupurine-treatments, that are dependent of the NUDT15 enzyme activity. So we performed a molecular epidemiology study in our population regarding the frequencies of deleterious alleles of this gene.

Second. As the authors discuss, it is relevant to know if the pathogenic variants of the admixed populations are related to the Native American ancestry. However, there is a lack of evidence to support this hypothesis and the authors should perform the additional analyses needed for a solid discussion. Ideally, the authors should use the whole exome data to accurately assess how pathogenic variants are correlated with Native American genetic global and local ancestry.

Third, in the interest of personalized medicine, it would be interesting to see if some Native American populations have higher frequencies of the pathogenic alleles than others. Similarly, the frequencies in the American admixed populations can be correlated or not to the average proportions of the Native American ancestry, depending if the Native American genetic sources were different and had different frequencies of the pathogenic alleles.

Therefore, assuming the authors only have access to the data regarding the NUDT15 gene, the authors should address the following analyses.

1) An extra table for the allele frequencies independently for each of the 11 Native American populations when the population size is five or higher. This table should also analyze independently the four American admixed populations of the 1000 genomes and include the southeastern Brazilian admixed population. The authors should use this information to confirm and discuss more accurately what they say in line 227.

Response: We accepted this suggestion. The supplementary table 2 contains data requested. The discussion section has been modified accordingly the results achieved by the analyzes performed (lines 281-288). 

2) A table reporting the FST between populations regarding all variants of the region analyzed.

Response: We have accepted the suggestion and created the table 4, placed in the Results section. 

3) If the authors do not have access to the whole exome data and can not compute the average genomic proportions of ancestral components (Native American, European and sub-Saharan African) for the American admixed populations, they should at least report the values found in the literature for these populations (the amazonian Brazilian admixed population, the southeastern Brazilian admixed population and the four American admixed populations) and see if the frequency of the studied variants is correlated with the Native American ancestry as the authors claim in line 241.

Response: We have added the information requested. (lines 305-317). 

• Authors must also make minor but important changes:

Introduction

1) Line 58: change ethnically by genetically

Response: The text has been modified accordingly.

Methods

2) Line 114: add detailed information of the location of sampling of the admixed and Native American populations

Response: We have added the information requested (Supplementary table 1).

3) Detail which regions are analyzed and in which chromosome they are found.

Results

Response: The regions are the chromosome location are detailed in table 1.

4) Sort tables 1 and 2 by position

Response: We have changed the tables in accordance with this suggestion. 

5) Paragraph starting in line 167 should be supported by a statistical test, like the Chi-squared test.

Response: We performed a Fisher’s exact test to support the differences in frequencies observed.

6) Correct the sentence of line 206 where it is said that Native American populations are descendants of the analyzed East Asian populations. Native American might be more closely related to East Asian populations than to other populations, but this does not mean they are their descendants.

Response: We have corrected the sentence in accordance with this suggestion.

---

## [Decision Letter · Decision Letter 1]

20 Feb 2020

PONE-D-19-30849R1

Identification of NUDT15 gene variants in Amazonian Amerindians and admixed individuals from northern Brazil

PLOS ONE

Dear Fernandes,

Thank you for submitting your manuscript to PLOS ONE. After careful consideration, we feel that it has merit but does not fully meet PLOS ONE’s publication criteria as it currently stands. Therefore, we invite you to submit a revised version of the manuscript that addresses the points raised during the review process.

We would appreciate receiving your revised manuscript by Apr 05 2020 11:59PM. To enhance the reproducibility of your results, we recommend that if applicable you deposit your laboratory protocols in protocols.io, where a protocol can be assigned its own identifier (DOI) such that it can be cited independently in the future. For instructions see: http://journals.plos.org/plosone/s/submission-guidelines#loc-laboratory-protocols

We look forward to receiving your revised manuscript.

Kind regards,

Francesc Calafell

Academic Editor

PLOS ONE

Reviewers' comments:

Reviewer's Responses to Questions

**Comments to the Author**

1. If the authors have adequately addressed your comments raised in a previous round of review and you feel that this manuscript is now acceptable for publication, you may indicate that here to bypass the “Comments to the Author” section, enter your conflict of interest statement in the “Confidential to Editor” section, and submit your "Accept" recommendation.

Reviewer #2: (No Response)

2. Is the manuscript technically sound, and do the data support the conclusions?

Reviewer #2: Partly

3. Has the statistical analysis been performed appropriately and rigorously? 

Reviewer #2: No

4. Have the authors made all data underlying the findings in their manuscript fully available?

Reviewer #2: Yes

5. Is the manuscript presented in an intelligible fashion and written in standard English?

Reviewer #2: Yes

6. Review Comments to the Author

Reviewer #2: The authors have addressed all the concerns I had pointed in the previous submission. However, there are few issues I think that can still be improved.

Line 36. Non-admixed should be written instead of pure

Line 123. genetic components should replace ethnic matrices

Line 177. The R-packages used should be listed and cited. I.e. the R packages used to compute the FST. If no package was used for the FST computation and a self-made script was used, the FST formula should be written and a citation to the reference article added.

Line 214 The p-values regarding the BAP population compared to the others should also be listed at Table 3. Moreover, the authors should apply a multi-test p-value correction. The p.adjust() function in the R package stats is one way to do it (although of course not the only one).

Line 234. FST values should not be described as percentages.

Line 237 BAP population must be included at table 4.

Line 261 Latin Americans with low percentages of Native American ancestry are not genetically similar to East Asians. In my opinion, this sentence is still not accurate. The authors can try a reformulation of the sentence to highlight that East Asians have the highest allele frequencies a part of Native Americans or directly replace exceeds 1% only in Asians and in populations genetically similar to them, that is, 262 the Latin and Amerindian populations by exceeds 1% only in Asians, Latin American and Amerindian populations.

Line 216. The text should be rewritten regarding the new table 3.

Line 281. The authors should avoid the expression we believe.

Line 296. The results shown in this paragraph do not match with table 2.

Line 302. The results shown in this paragraph do not match with table 2.

Line 312. The authors should cite a paper where CLM, PUR, MXL and PEL populations are analyzed all together. Moreover, some of the articles cited have analyzed similar populations but not the 1000 genomes. The following articles perform local ancestry analyses of CLM, PUR, MXL and PEL populations, while the last one, also includes local ancestry information for a South-Eastern Brazilian population. https://journals.plos.org/plosgenetics/article?id=10.1371/journal.pgen.1004023

https://www.cell.com/ajhg/fulltext/S0002-9297(17)30107-6

https://www.nature.com/articles/s41598-019-50362-2

Line 315 To be able to state “demonstrate” the authors should carry out a correlation test. Either the text should be adapted or the correlation performed.

Supplementary table 1 is a presentation of the new data generated by the authors.

Thus, the BAP population should be included (differentiated of the Native Americans), and therefore the second column can be renamed as population.

Supplementary table 2 should also include BAP population.

Finally a more coherent nomenclature can be used. I.e. a unified criteria to use the rs ID or the aminoacid subsitution. Moreover, the authors should reference the tables in the text when they discuss them explicitly.

To conclude, the genetic information of the BAP population should also be available, together with the Native Americans.

7. PLOS authors have the option to publish the peer review history of their article (what does this mean?). If published, this will include your full peer review and any attached files.

Reviewer #2: No

---

## [Author Response · Author response to Decision Letter 1]

26 Mar 2020

We are hereby submitting a new revised version of the Original Article entitled “Identification of NUDT15 gene variants in Amazonian Amerindians and admixed individuals from northern Brazil”, submitted previously as manuscript PONE-D-19-30849R1. We would like to thank the Reviewer for their high quality and constructive reviews that greatly helped improved our manuscript, and the Editor for his careful reading.

In this revised version of the manuscript, we did our best to address the new comments raised by the Reviewer 2 (all the major modifications are marked in green during the file “Revised Manuscript with Track Changes”). Additionally, we have no legal or ethical restrictions on sharing data publicly. Therefore, we have uploaded the minimal anonymized data set necessary to replicate our study findings as a Supporting Information file, named “Anonymised datasets”.

A detailed item-by-item response to each of the Reviewer’s points has been uploaded in the PLOS one submission page. Our replies are marked in italic.

Given the above, we hope that your suggestions have been adressed. Thank you for the attention.

---

## [Editor Report · Decision Letter 2]

30 Mar 2020

Identification of NUDT15 gene variants in Amazonian Amerindians and admixed individuals from northern Brazil

PONE-D-19-30849R2

Dear Dr. Fernandes,

We are pleased to inform you that your manuscript has been judged scientifically suitable for publication and will be formally accepted for publication once it complies with all outstanding technical requirements.

With kind regards,

Francesc Calafell

Academic Editor

PLOS ONE
---

## [Editor Report · Acceptance letter]

1 Apr 2020

PONE-D-19-30849R2 

Identification of *NUDT15* gene variants in Amazonian Amerindians and admixed individuals from northern Brazil 

Dear Dr. Fernandes:

I am pleased to inform you that your manuscript has been deemed suitable for publication in PLOS ONE. Congratulations! Your manuscript is now with our production department. 

With kind regards,

on behalf of

Dr. Francesc Calafell 

Academic Editor

PLOS ONE